

# Designing online species identification tools for biological recording: the impact on data quality and citizen science learning

Nirwan Sharma[1,2], Laura Colucci-Gray[3,4], Advaith Siddharthan[5],
Richard Comont[6] and René van der Wal[2]

[1] School of Natural and Computing Sciences, University of Aberdeen, Aberdeen, UK
[2] School of Biological Sciences, University of Aberdeen, Aberdeen, UK
[3] School of Education, University of Aberdeen, Aberdeen, UK
[4] Moray House School of Education, University of Edinburgh, Edinburgh, UK
[5] Knowledge Media Institute, The Open University, Milton Keynes, UK
[6] Bumblebee Conservation Trust, Stirling, UK

Corresponding author
Nirwan Sharma,
n.sharma@abdn.ac.uk

## ABSTRACT

In recent years, the number and scale of environmental citizen science programmes that involve lay people in scientific research have increased rapidly. Many of these initiatives are concerned with the recording and identification of species, processes which are increasingly mediated through digital interfaces. Here, we address the growing need to understand the particular role of digital identification tools, both in generating scientific data and in supporting learning by lay people engaged in citizen science activities pertaining to biological recording communities. Starting from two well-known identification tools, namely identification keys and field guides, this study focuses on the decision-making and quality of learning processes underlying species identification tasks, by comparing three digital interfaces designed to identify bumblebee species. The three interfaces varied with respect to whether species were directly compared or filtered by matching on visual features; and whether the order of filters was directed by the interface or a user-driven open choice. A concurrent mixed-methods approach was adopted to compare how these different interfaces affected the ability of participants to make correct and quick species identifications, and to better understand how participants learned through using these interfaces. We found that the accuracy of identification and quality of learning were dependent upon the interface type, the difficulty of the specimen on the image being identified and the interaction between interface type and 'image difficulty'. Specifically, interfaces based on filtering outperformed those based on direct visual comparison across all metrics, and an open choice of filters led to higher accuracy than the interface that directed the filtering. Our results have direct implications for the design of online identification technologies for biological recording, irrespective of whether the goal is to collect higher quality citizen science data, or to support user learning and engagement in these communities of practice.

## INTRODUCTION

The relationship between science and society and the existing knowledge divide between expert and lay knowledge continues to be subject of extensive debate (*Mooney, Duraiappah & Larigauderie, 2013*; *Deshpande, Bhosale & Londhe, 2017*). Citizen science as originally introduced by *Irwin (1995)* is a process of active engagement of the public in matters pertaining to science and technology, with a view to influence policy developments. Citizen science has also been associated with 'public participation in scientific research' (*Cohn, 2008*; *Bonney et al., 2009*). Participation in such initiatives is meant to bridge the expert/lay knowledge divide (*Hage, Leroy & Petersen, 2010*), in the hope that by engaging the general public with scientific research, society will benefit too (*Riesch & Potter, 2014*).

The rapid advancement of computing technologies—especially mobile computing and the Internet—has led to the emergence of a large number of citizen science projects in a wide range of domains, including astronomy (e.g. classifying shapes of galaxies, (*Raddick et al., 2010*)), medical sciences (e.g. contributions to protein engineering for drug discovery; *Cooper et al., 2010* and cancer diagnostics; *Schrope, 2013*) and environmental sciences (e.g. obtaining biological records through identification of plant or animal species on images captured by camera traps; *Swanson et al., 2016* or by volunteers; *Silvertown et al., 2015*). Through initiatives of this kind, digital technologies have created new opportunities for engagement by a much wider range of people with both the products and processes of scientific research. Hence, the role of the public has changed from being simple 'recipients' of scientific developments to acting as contributors to research, for example, by helping to collect and categorise data for scientific projects (*Silvertown, 2009*).

While such new opportunities offer the prospect of scaling up scientific research activities, the inclusion of lay people into the activities of professional scientists is not without problems. Two perennial concerns reported in the literature are volunteer retention and differential ability (*Conrad & Hilchey, 2011*; *Van der Wal et al., 2016*; *Austen et al., 2016*, *2018*), also described—by *Collins & Evans (2002)*—as the 'dilemma of experience and expertise'. As the number of stakeholders involved in research expands, digital tools may facilitate a corresponding shift from a 'contributory expertise' model of engagement, based on the delivery of data or expert knowledge, to an 'interactional expertise' model, based on the integration of lay and expert knowledge to form 'interactional expertise' (*Collins & Evans, 2002*). The difference between the two types of expertise—contributory and interactional—lies with the degree of stakeholders' immersion in core research practices. Whilst contributory expertise is based on the provision of data and/or information to and from the experts, interactional expertise presupposes the ability to act in a specialist domain of research, and thus interact with specialists even 'in the absence of practical competence' (*Collins & Evans, 2002*). By definition, interactional expertise relies heavily on ongoing interaction with experts, and quickly becomes out of date when interaction is not sustained. Two aspects are therefore paramount: (i) access to specialist knowledge for sound decision-making,

and (ii) level of sustained interaction with experts. Both aspects are significantly affected by digital technologies. Thus, understanding the potential—and pitfalls—of new digital interfaces is urgently required in order to maximise the benefits for citizen science (*Kahn, Severson & Ruckert, 2009*). In this context, we study the differential affordances of alternative digital interfaces for biological recording, whereby the interface acts as the first realm of development of interactional expertise.

## Context of the study

This study is embedded in the context of biological recording. Heavily dependent upon large-scale data collection with wide geographical scale, biological recording relies on incremental and cumulative data gathering. Traditionally, biological records have been assembled by field naturalists—on their own accord and as part of natural history societies and recording clubs (*Burnett, Copp & Harding, 1995*; *Miller-Rushing, Primack & Bonney, 2012*), well before the practice was set to become part of an academic discipline, and greatly preceding the advent of digital technologies. Species identification skills were developed as part of 'field' immersion, through familiarity with the context, sharing of annotations, conversations with local people, and direct observation. Key to identification was thus the combination of formal and tacit/experiential knowledge acquired and maintained by the experienced recorder. Conversely, citizen science projects extend traditional roles to include non-experts, that is, people who may have an interest in a topic but lack direct, relevant experience of field-note taking, recording or identification. In addition, and as indicated above, digital tools enable geographical extension, thus offering the possibility to answer the pressing needs for collecting species distribution data, while creating 'virtual' scenarios in which large-scale data gathering occurs by decoupling people from places, short-cutting the long-held experience and tacit knowledge of the field recorder. So, differently from biological recording occuring through the employment of traditional, experiential expertise, digital tools simulate the process of species identification via engagement with different types of interactive interfaces. In order to understand the function of digital tools in biological recording and their effective design and use, there is thus a need to investigate their role in creating communities of practice (*Lave & Wenger, 1991*), whereby participants with different levels of training and different cognitive abilities deliberate and debate, thereby creating opportunities to learn and think together (*Bonney et al., 2009*; *Dickinson, Zuckerberg & Bonter, 2010*). Communities of practice may range in size and membership; and interaction amongst members may be in real-time, off-line, or facilitated by a virtual environment. Importantly, communities of practice are actively orientated towards enhancing participants' learning practices through request for information, problem solving, coordination, or seeking advice and experience (*Lave & Wenger, 1991*). Within the realm of a community of practice, digital interfaces can be conceived as 'tools' (*Adedoyin, 2016*) for facilitating interaction, and evaluated from a user perspective on the basis of their 'suitability for use' (e.g. the extent to which they align with users' psychological inclinations and offer accurate as well as appropriate information), 'accessibility' (e.g. clarity and immediacy of use) and 'opportunity for learning' (e.g. availability to revisit prior steps and adapt new

learning). In the context of biological recording, digital technologies would thus enable, and in some way re-create, the ongoing 'back and forth' of identification processes (e.g. looking for features; comparing with previous knowledge, images, or experience) and intersection of different types of knowledge which are held by different groups at different levels of immersion in the practice of biological recording. Participants rely on learning support provided by identification tools, such as identification keys and field guides, sometimes augmented through direct training from other, more experienced, members or participation in community events. Through such processes, participants face challenges deriving from the need to make sense of new information, but they can also develop expertise such as handling scientific language for species identification. In this view, two key aspects emerge as central to the focus of this study, and we concentrate on these as a basis for designing interfaces which can favour progressively more extended interactional opportunities. First, the design of interfaces needs to take account of both communication formats and accuracy of identification tools in order to support participation from different types of users. This aspect was investigated quantitatively by measuring user performance and cognitive workload associated with exposure to new information. Second, interfaces need to take account of the opportunities for participants to develop their own learning and thus increase interactional expertise. This aspect was looked at qualitatively through appraising the 'suitability of use' of digital tools, for example by helping with the recognition of new terms in practice and/or revising choices made during the process.

## Designing digital identification tools

Two common tools for species identification are keys and field guides. Identification keys provide novices with access to specialist taxonomic knowledge; yet, as *Lobanov (2003)* pinpoints, they are notorious for their difficulty of use: 'Keys are compiled by those who do not need them for those who cannot use them'. This has led to increasing calls for more user-friendly identification keys to foster their adoption and use (*Stevenson, Haber & Morris, 2003*; *Walter & Winterton, 2007*).

By contrast, field guides are usually popular tools for the casual consumers of taxonomic information, and typically contain information derived from identification keys (*Stevenson, Haber & Morris, 2003*). Field guides exist in various forms, such as books, posters, flashcards and brochures, and are often easier to use than identification keys (*Scharf, 2008*). Unlike keys, field guides do not have as strict a design structure, and the presentation of information can therefore vary considerably. They usually contain illustrations or photographs together with written descriptions to aid species identification. Field guides may also include simple keys/glossaries which can be presented as illustrations.

According to the theory of situated learning and legitimate peripheral participation in communities of practice, any form of learning is contextual to the social setting where it is practiced (*Lave & Wenger, 1991*), and thus usability also depends upon the 'context of use' of an interface (*ISO 9241-11, 1998*). In this view, both types of identification tools, keys and guides, may be assumed to be designed for the respective social setting of
intended use (*Hung, 2002*; *Stevenson, Haber & Morris, 2003*), namely 'the lab', for identification keys and 'the field' for field guides (*Stevenson, Haber & Morris, 2003*). It can be argued that both identification keys and field guides ought to be 'fit for their purpose', as they have been widely used by identification experts for hundreds of years (*Scharf, 2008*). As we will see later, features of both tools were incorporated in the design of interfaces—and suitably evaluated—in order to address a wider set of identification processes, and assess how the use of digital interfaces can be maximised for different types of users.

## Online user participation

As indicated earlier, the availability of portable computing technologies, such as mobile phones, digital photography and automated location tagging through GPS, has notably extended the opportunities for biological recording. Participants can contribute species data, often captured through digital photographs, and there is growing evidence of this in media sharing portals (e.g. Flickr, Facebook, YouTube) as well as in citizen science projects (e.g. iSpot, iNaturalist, eBird), and these are increasingly being mined for biodiversity research (*Sullivan et al., 2009*; *Winterton, Guek & Brooks, 2012*; *Gonella, Rivadavia & Fleischmann, 2015*; *Silvertown et al., 2015*). However, while online identification tools (keys/guides) are burgeoning, there is a surprising lack of user-centred design studies to inform their development. *Stevenson, Haber & Morris (2003)* reasoned that new Electronic Field Guides should be developed using a combination of field guides and identification keys, with potential applications to citizen science projects. The authors also highlighted the application of learning theories and knowledge representation techniques in the design of these tools, as well as the importance of testing them with end users as an essential requirement of any interactive software to improve usability (*Stevenson, Haber & Morris, 2003*). These principles are likely good starting points for research into the design of digital identification technologies aimed at enhancing expertise of citizen science project users, and so were adopted in this study detailed as follows.

## User-centred design for new online identification technologies

All identification keys work on the basis of elimination, by filtering out possible species based on observed attributes. This is less prone to error than the method of direct matching of attributes used in field guides, especially for novices identifying unknown or unfamiliar specimens (*Wills, Inkster & Milton, 2015*). Visual classification begins with the identification of individual attributes of a specimen, which are subsequently combined and weighted (e.g. in the mind, or by using a key or guide) to reach a decision (*Treisman & Gelade, 1980*; *Thompson & Massaro, 1989*; *Lamberts, 1995*; *Wills, Inkster & Milton, 2015*). So, if users are identifying new, multi-attribute objects, like animal or plant species, direct matching may lead to misidentification because of the high likelihood of decision-making based on irrelevant features (*Wills, Inkster & Milton, 2015*). Moreover, directing user attention—a distinctly limited resource (*Lavie, 1995*; *Logan, 2002*)—to relevant attributes and distinguishing features that aid identification

**Table 1 Comparison of characteristics of the three different identification tools evaluated in this study (field guide, feature selection and decision tree).**

| Characteristics | Field guide (Control) | Feature selection | Decision tree |
|---|---|---|---|
| Type of identification key | Paper-based single access (dichotomous/polytomous) | Interactive multi-access | Interactive single access, (dichotomous/polytomous) |
| Order of decision-making | Partitioning species into biologically-informed subcategories | Open-choice selection of visual features | Directed by interface: easy visual features decided first, and harder features later on |
| Identification mode | Visual comparison of all species | Interactive filtering out of species that do not match selected features | Interactive filtering out of species that do not match selected features |

(*Gibbon, Bindemann & Roberts, 2015*) by highlighting key differentiation attributes, and using non-technical language and representative pictures, becomes particularly important for novices who may be inexperienced in identifying which attributes are relevant for a particular species identification.

For experts, information processing of stimuli takes place in the working memory (*Baddeley, 1992*), where they can access prior knowledge from their long-term memory, extract species attributes and use that to arrive at an identification. Novices without in-depth species knowledge rely on some form of training, which in online citizen science projects is typically delivered through an identification tool. As such, identification tools are designed for use on computing devices, and well defined deterministic procedures or rules exist for species identification (though subject to change over time with the discovery of new species, or new variations within species); species identification for novice users can therefore be modelled as a structured problem in interface design (*Jonassen, 2000*). Taking such principles into account, this study first proceeded by considering the methods of elimination using identification keys, and using illustrations and language as well as contextual information derived from a field guide with progressive disclosure (showing information when it may be needed). Specifically, we prototyped two interactive interfaces—'Feature selection' and 'Decision tree'—the building blocks of which were taken from an existing 'Field guide', the latter in turn acting as control. These interfaces vary with respect to several generic characteristics of identification keys: the number of entry points (single or multi access) and number of states (dichotomous (two states) or polytomous (more than two states) (*Lobanov, 2003*)); whether the user or the interface directs decision-making; and whether the interface supports filtering. See Table 1 for a summary of the key differences between the prototyped interfaces.

In the remaining part of this paper, we report on a systematic evaluation of the two interactive prototypes (and their associated field guide), with a sample of semi-experienced users, focussing on two key aspects: (a) decision-making—how the accuracy of identification, time taken and mental workload of a user is affected by the choice of interface; and (b) quality of learning process—how the different interfaces facilitate novices in sophisticated thought processes that require the combination of generic prior skills (e.g. image recognition, colour and size perception, language processing), with task-specific decision-making moderated by the interfaces (as described in Table 1).

Such comparative assessment would provide central information on design principles and the accuracy of learning pathways enabled by each interface type.

## MATERIALS AND METHODS

### Focal species group

We selected bumblebee species in the UK as the taxonomic focus of this study. Bumblebees represent a species group which requires careful differentiation of visual features for accurate identification. There are 22 species (*Siddharthan et al., 2016*) in the UK that are identifiable based on different visual features, including the colour of the tail, size of the face, shape of antennas, different colour band patterns on their thorax and abdomen, and presence of pollen baskets on the hind legs. Combinations of these features assist the classification of bumblebees into different species as well as to identify a specimen as female (queen/worker) or male. Not all features are clearly visible in all species or in all circumstances, and this is potentially a bigger issue when performing identification using a photograph rather than in the field. Some of the species are very similar; for example, only close inspection of the colour of the hairs on the hind legs can distinguish a Red-tailed bumblebee (*Bombus lapidarius*) from a Red-shanked carder bee (*Bombus ruderarius*).

### Images

We randomly selected six images from a real world citizen science project, BeeWatch (*Van der Wal et al., 2015*), with two levels of difficulty due to some bumblebee species being intrinsically harder to identify than others. Those images were selected from a large subset for which BeeWatch had solicited 10 independent identifications by BeeWatch users, which was used to develop a Bayesian verification algorithm (*Siddharthan et al., 2016*). All images were also identified by a bumblebee expert and we selected three difficult images where fewer than 50% of users identified the species correctly (compared to the expert), and three easy images where more than 80% of users identified the species correctly.

### Interfaces

The three interfaces used in the study and described in Table 1 are illustrated in Figs. 1–3. In the UK, the Bumblebee Conservation Trust (bumblebeeconservation.org) provides a bumblebee identification guide, designed using relatively simple language and containing illustrations of bumblebee species, with the aim to help members of the general public to perform the task of bumblebee identification in the field (see Fig. 1). We used the images and textual information from this guide to develop an interface employing the principles of decision trees, shown in Fig. 2 with an example workflow of an identification process. Each screen asks the user to select an option in a predetermined order, such that cognitively easier options are offered early on. Users can backtrack on a decision by following the 'back' link at any stage. Using the same images and textual information, we also designed a tool employing feature selection (Fig. 3). Users can select any features they are confident about, and their selection will shade out options that do not match. Features can be unselected, or their values changed at any point.

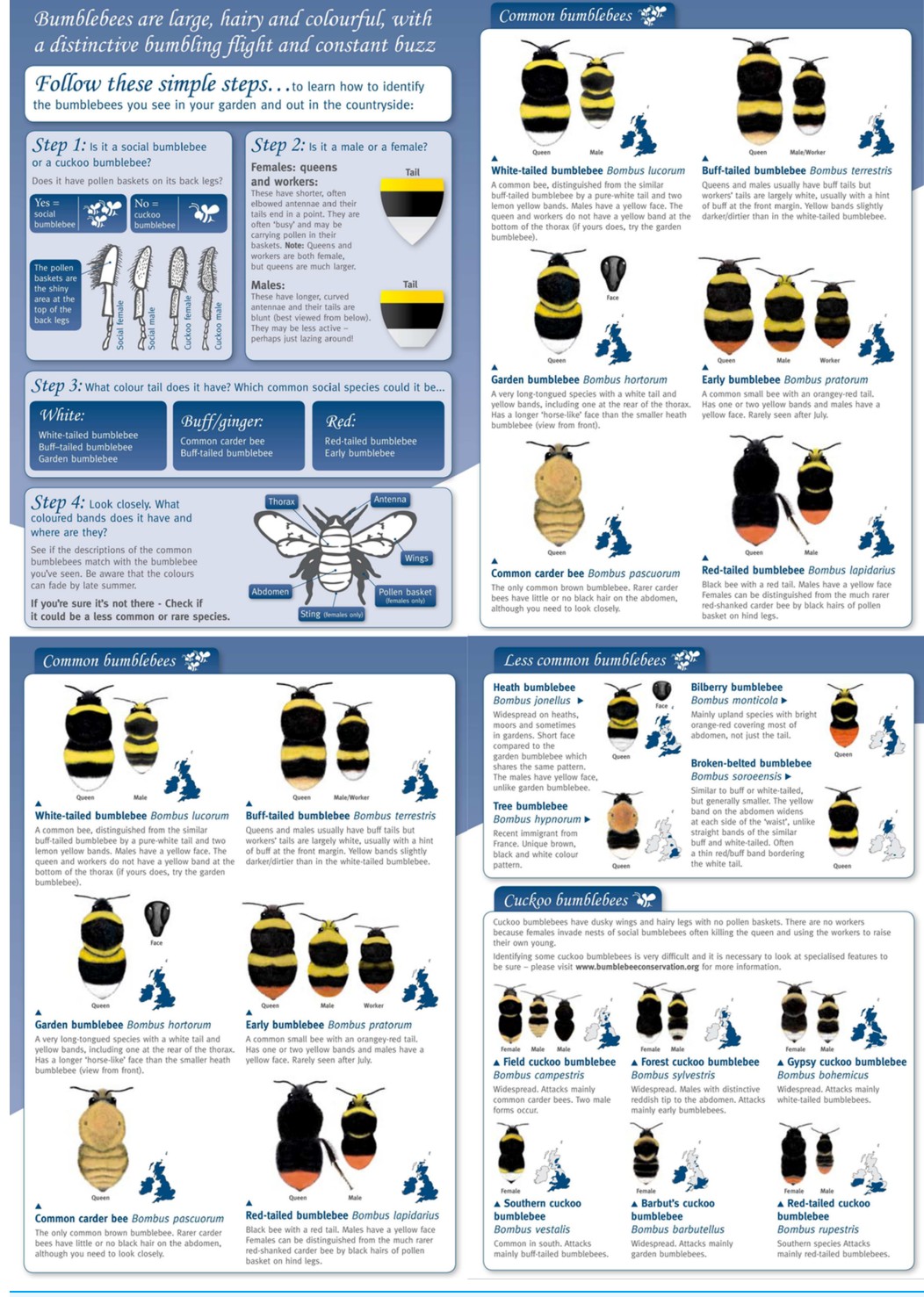

**Figure 1 Field guide.** Source: Bumblebee Conservation Trust (http://bumblebeeconservation.org).

The interfaces studied incorporate design principles from both Field guides and Identification keys. The first set of principles considered the 'context of use' of the interfaces with associated behavioural, aesthetic and anatomical characteristics.

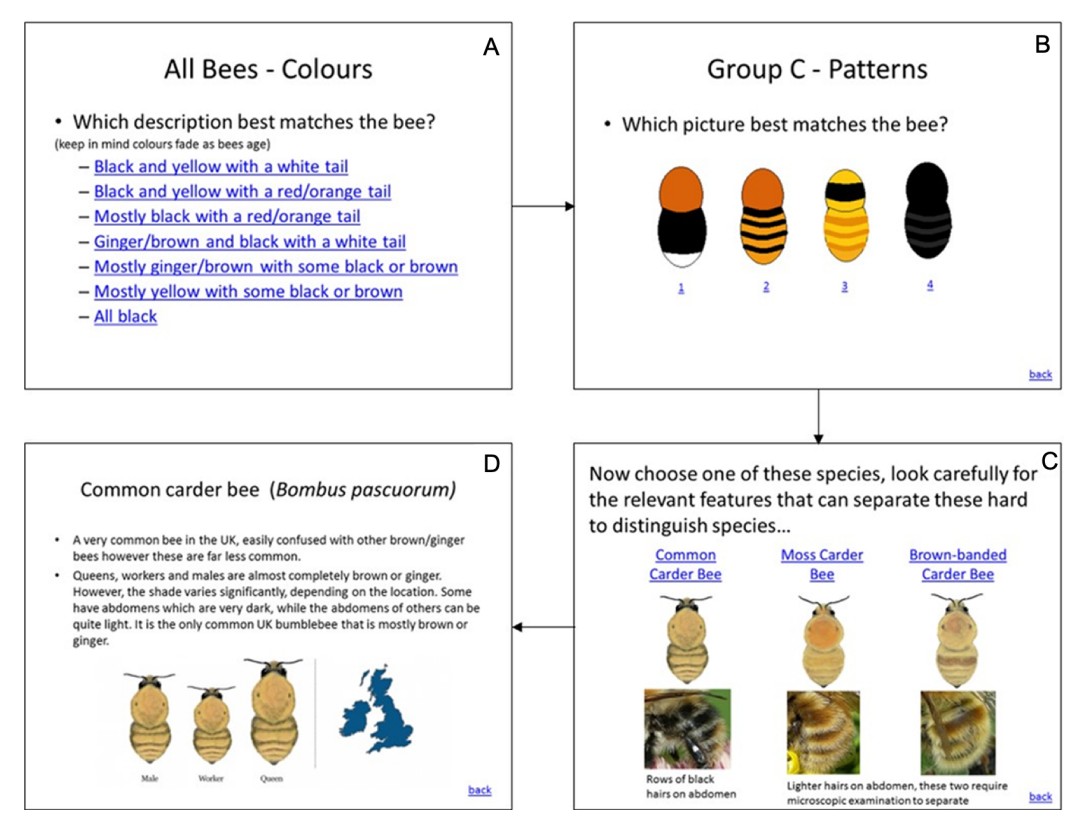

**Figure 2 Decision tree tool.** Workflow from (A) to (D). The order of selections is 'Mostly ginger/brown with some black or brown' in (A), option '2' in (B) and 'Common Carder Bee' in (C).

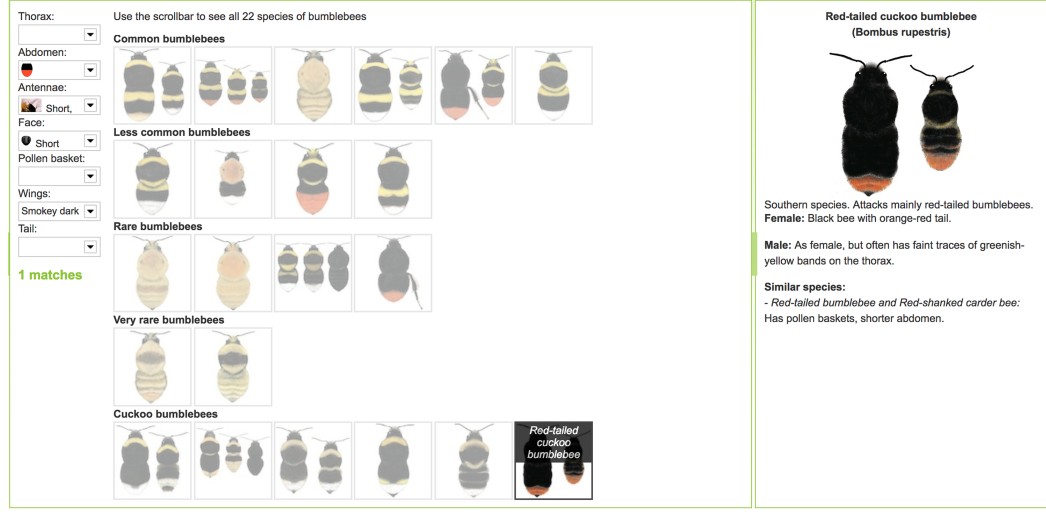

**Figure 3 Feature selection tool.** When activating drop-down filters, all species not corresponding with the choices made are 'shaded out'. In this specific example, the respective filter settings for 'Abdomen', 'Antennae', 'Face' and 'Wings' shade out all but the Red-tailed cuckoo bumblebee. A more detailed description of the resulting species is then provided.

The second set of principles considered a hierarchy of significant features which are deemed to be 'distinctive' for particular species. Hence, the design of the interface considered both holistic and analytical functions of the identification process. In addition, our study involved a species group which is neither too simple (i.e. few species with easy to identify characteristics) or too complicated (i.e. a species group such as moths for which ~2,500 species are known to occur in the UK) to identify. The developed interfaces had design parallels in other species identification tools. For instance, the Decision tree interface has design elements (single access key design) similar to the Merlin Bird ID mobile application (http://merlin.allaboutbirds.org/) and the Natural History Museum Bumblebee ID key (http://www.nhm.ac.uk/research-curation/research/projects/bombus/key_british_colour.html), while the Feature selection interface has design elements of multi-access keys such as interactive keys on iSpot (www.ispotnature.org/webkeys/index.jsp) and Discover Life (www.discoverlife.org). Some citizen science projects even provide multiple types of interactive tools, such as Go Botany (https://gobotany.newenglandwild.org), which offers a dichotomous key as well as a multi-access key based design (*Farnsworth et al., 2013*). For the purpose of this study, we looked more closely at design choices in relation to data quality and user-learning, thus contributing empirical findings to this growing area of practice.

## Participants

Participants were drawn from students on the MSc Ecology and MSc Applied Marine and Fisheries Ecology degrees at the University of Aberdeen. Participation was couched as an opportunity for wider learning on citizen science and to contribute to ongoing research. All participants offered their voluntary participation, showing self-selection, and took part in fixed 30–40 min slots with the experimenter (the lead author). Each participant was given £10 in cash as a token of our appreciation for their participation. In addition, a collective session was arranged with the participants following the experiment, to highlight the results of the exercise and introduce them to the role of citizen science in nature conservation. The sample size was limited to 18 participants to keep the data manageable (108 identification tasks with think-aloud and screen recording data) (*Anderson & Vingrys, 2001*). The study was approved by the University of Aberdeen's ethics committee, and all participants provided written informed consent.

Participants differed in their level of English language skills (seven international students of which two from English-speaking countries, and a further 11 native English speakers from the UK). The median age of participants was 23 years. Eight participants were female and the other 10 were male. Almost all had some experience with species identification of plants or animals other than insects (due to exposure to this during their biology degrees), while 13 participants also had some experience with insect identification. None of them were experienced in bumblebee species identification and therefore could all be considered novices for this species group.

Masters level students are routinely exposed to online systems like digital libraries (Ebrary), databases (Scopus, Web of Science) and learning environments (Blackboard,

used by all University of Aberdeen students) and regularly use digital systems for their assignments, tutorials and presentations, including word processors, spreadsheets, graphical software (notably Powerpoint) and statistical software. We therefore expected our sample of users to be able to provide insights into the role played by a respective interface in reaching a (positive) species identification, as well as feedback on the role of the interface in mediating their cognitive load.

## Experimental design

We used a concurrent mixed methods design involving bumblebee identification tasks, workload questionnaires, think-aloud protocol, and screen recordings. Before commencing, participants were given a brief background to the study, informing them that it involved completing bumblebee identification tasks using three different interfaces shown on a computer in order to assess the usability and effectiveness of these interfaces. All the participants then completed six identification tasks. Each task concerned the use of one of the three interfaces and one of the six photographs of bumblebees used in the study (and thus six photographs per participant), in a manner counterbalanced across participants to control for order of identification tasks (see Appendix 1) whilst ensuring all interface types were used (twice) by each participant.

The participants were given a practice run with each interface when they encountered it for the first time during the experiment. The following steps were then performed for each identification task:

1. The participant was given an image of a bumblebee printed at high resolution on paper.
2. An identification interface was shown on a laptop, with a mouse attached for easier interaction.
3. The participant was asked to identify the bumblebee species in the image by using the interface and to subsequently write down the identification on an answer sheet. The identification process was timed, but it was stressed that the goal was to get an accurate result, and thus to take as much time as required.
4. During the identification process the participant was asked to think aloud and this was recorded.
5. A paper version of the NASA-Task Load Index (NASA-TLX) subjective workload questionnaire (see below) was filled out by the participant for the completed identification task.

## Experimental measures

We used quantitative methods to assess accuracy, time taken and cognitive workload, and a think-aloud protocol to collect qualitative data.

### Accuracy

Task accuracy was assessed by comparing the identification provided by the participant to the expert identification (provided by a Bumblebee expert through the BeeWatch platform).

### Time

Task time was the total time taken to complete a single identification task (i.e. one photo on one interface) measured in seconds.

### Workload

As identification is a problem-solving task, we wanted to obtain relative measures of 'how much mental work' was required by the participants to perform the task. We used the NASA-TLX subjective workload questionnaire (Appendix 2), a workload assessment tool (*Hart & Staveland, 1988*) that has been widely used for assessing workload measures for Human Computer Interaction research (*Hart, 2006*). The assessment tool provides a weighted average across six dimensions of mental workload: Mental demand, Physical demand, Temporal demand, Performance, Effort and Frustration. In the first part of the questionnaire, participants give—for each of the six dimensions—a subjective rating (scale 0–100) per identification task. In the second part, they assign weights to the dimensions by conducting a pairwise comparison for all possible combinations of the six dimensions. For each pairwise comparison, participants select which of the two dimensions is perceived to be the most important contributor towards the workload for the task. The selections are then used to assign weights from 0 to 5 for each dimension, by counting the number of times each dimension is selected as being the most important across all pairwise comparisons. For example, if Mental demand is selected as most important in each of the five possible pairwise comparisons, then it is assigned a weight of five. These weights are then multiplied with their associated ratings and added to derive at the total workload score, which is subsequently divided by 15 (total sum of the weights) to obtain an average workload score for each task. The participants were explained the NASA-TLX rating scale and its six dimensions after the first task, and were also provided with a 'workload information sheet' for reference.

### Think aloud

Participants were asked to think aloud and verbalise their thought processes while performing the identification task (*Ericsson & Simon, 1980*) and this was recorded. Think aloud is a protocol widely used in research concerned with learning processes (*Leow & Morgan-Short, 2004*) as well as usability studies (*McDonald, Edwards & Zhao, 2012*), and is found to be highly suited for capturing user thought processes without negatively affecting mental workload or performance measures (*Young, 2005*; *Fox, Ericsson & Best, 2011*; *Pike et al., 2014*). Some of the issues of the think-aloud method, such as reactivity, verbal ability of participants and validity, were addressed by following standardised approaches from the literature. These included giving a practice run on the tasks, employing 'keep talking' probes, capturing additional data in the form of screen recordings, and using a research task which is not 'automatic' for the participants (*Young, 2005*). The experimenter did not interact during the identification process; however, if participants fell silent during the task, they were asked to resume talking.

Data was analysed using grounded theory (*Wagner, Glaser & Strauss, 1968*) by firstly coding the data to identify patterns of use of the interfaces, with each observation

annotated with the relevant image and interface. These patterns of interface use helped in identifying the most commonly followed steps taken by participants during identification tasks across each interface and image combination. Further, 'suitability of use' of each interface was determined by investigating three dimensions, derived from insights provided by the literature (*Adedoyin, 2016*), namely *usefulness* (i.e. whether the interface facilitated progress or not), *accessibility* (ease of use in locating information on the interface) and *opportunity* for *learning* (mobilising prior and extended knowledge). These annotations allowed us to not only obtain information on the interactions with the interface but, most importantly, to also reveal the thought processes behind those interactions. This included looking at the identification strategies adopted by participants (e.g. colour matching vs. evidence-based reasoning) and the opportunities offered by the interface to verify possible mistakes (e.g. by going back) and to combine additional or new options (e.g. colour and position).

### Statistical approach

Quantitative analyses were run in SPSS Version 24 (SPSS Inc., Armonk, NY, USA). Accuracy, Time and Workload data were analysed using generalised linear mixed models. Accuracy data were fitted using a binomial distribution with logit function, while Time and Workload data were fitted using normal distributions with identity function. Image difficulty (Easy and Difficult), Interface type (Field guide, Feature selection, and Decision tree) and the interaction between them were fitted as fixed effects, and Participant was included as a random effect. Post-hoc comparisons were computed (using EMMEANS TABLES) for the interaction of Image difficulty and Interface type with least significance difference adjustment.

## RESULTS

### Accuracy

Our first hypothesis (H1) was that species identification accuracy is influenced by interface design, with interactive keys resulting in more accurate identifications than a field guide design, and with easier images resulting in higher accuracy than difficult images regardless the interface used.

Interface type ($F_{2,102} = 3.88$, $p < 0.05$), Image difficulty ($F_{1,102} = 5.02$, $p < 0.05$) and the interaction between Interface type and Image difficulty ($F_{2,102} = 4.39$, $p < 0.05$) all explained significant amounts of variation in accuracy of participants' species identification (Fig. 4A), revealing that it was not simply the type of interface or image difficulty alone that influenced identification accuracy, but it was their combination that mattered. Overall, our participants indeed achieved lower average accuracy with difficult images (mean accuracy = 0.24) compared to easy images (mean accuracy = 0.48). Post-hoc analysis, conducted to interpret the interaction, showed that for easier images participants achieved significantly higher accuracy using the Feature selection interface compared to the Field guide ($t_{102} = 2.26$, $p < 0.05$) and Decision tree ($t_{102} = 4.29$, $p < 0.0001$) interfaces. For difficult images, the Decision tree interface was significantly better than the Field guide ($t_{102} = 2.01$, $p < 0.05$). No other contrasts (within either category of image difficulty) were significant.

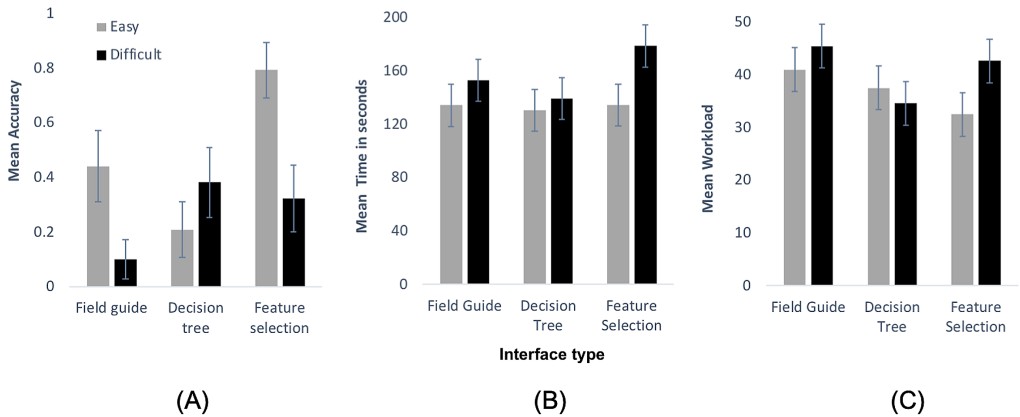

**Figure 4** **Quantitative analysis graphs.** (A) Mean (± SE) accuracy (0–1), (B) mean time taken (in sec) and (C) mean workload scores (scale 0–100) for each of the three studied interface types, for easy (grey bars) and difficult images (black bars) separately.

## Time taken

Our second hypothesis (H2) was that the time taken for species identification differs among the three focal interfaces and between image difficulty types (easy/difficult), with the decision tree (that orders questions by difficulty) taking the shortest time and the field guide (that offers no interaction) taking the longest, and with easier images taking less time than difficult ones.

Contrary to our expectation (H2), the type of interface did not influence the length of time it took participants to complete an identification task ($F_{2,102} = 1.21$, $p = 0.30$). Image difficulty mattered, however ($F_{1,102} = 4.43$, $p < 0.05$), with easier images (mean 133 s) taking less time to identify than difficult ones (mean 157 s).

## Workload

Our third hypothesis (H3) was that the perceived workload differs among the three interfaces and between the two image types, in ways similar to H2, with the decision tree being easiest to use, as it asks questions in order of difficulty and guides the user to an answer, and the non-interactive field guide being the hardest.

Our analysis of workload suggested differences between the three Interface types ($F_{2,102} = 3.81$, $p < 0.05$), but also gave some support for the notion that no one interface generated the lowest workload, but that this depended on Image difficulty (Interface type × Image difficulty: $F_{2,102} = 2.92$, $p = 0.06$; Image difficulty: $F_{1,102} = 3.09$, $p = 0.08$). Post-hoc analysis revealed that for easier images participants perceived significantly lower workload using the Feature selection tool compared to Field guide ($t_{102} = 2.21$, $p < 0.05$). For difficult images, participants perceived significantly lower workload for the Decision tree tool compared to both the Field guide ($t_{102} = 2.81$, $p < 0.05$) and Feature selection ($t_{102} = 2.09$, $p < 0.05$). No other contrast statements (within category of difficulty) revealed significant differences.

## Think aloud

While our quantitative data partially confirmed our hypotheses (H1–H3), they also produced several unexpected findings. Our qualitative data were used to subsequently help

interpret the above-reported findings in greater depth by looking at the nature of the thinking and learning processes.

### Field guide

Because interactive filtering—which is normally used to limit the number of relevant species to consider—could not be done with the field guide, participants tended to rely on matching colour patterns first, as a means to identify a bumblebee specimen. Visually, these appeared to be the most salient features. This strategy only proved useful in cases where the colour pattern was (a) identified correctly and (b) where it reduced the candidate species to a small subset, such as for participant 16 (“*colour of the tail black, so it's all black—we will go through the pictures*”) and participant 10, who went through the guide to determine which ones had a red tail—a rather specific feature (“. . . *maybe it's a red tailed bumblebee as looks more like that sort of . . .* ”), then said “*I quite like that one*”, to then make a resolute (and correct) choice.

Yet, in many cases there appeared to be too many species with similar overall appearances to allow functional matching. Participants experienced frustration due to lack of filtering, as stated by participant 1 (“. . . *I don't know how to narrow it down more based on the pictures . . .* ”). When some motivated participants subsequently tried to identify a specimen based on other attributes (e.g. pollen baskets, distinguishing attributes), the total number of attributes to be considered increased, often beyond the capacity of participants to process information in their minds (i.e. exceeding working memory). This repeatedly led to ‘guesses’ whilst overlooking attributes critical to the identification. For instance, participant 6, after poring over numerous different attributes (colour pattern, pollen basket, wing colour, yellow band at the bottom of the thorax, size of face), specifically considered the correct category of cuckoo bumblebees (“*it's got just dusky wings so it could be a cuckoo bumblebee*”), but then dropped this understanding, focussed on other, less useful attributes for this particular identification (“*It does have yellow band so garden bumblebee*”). This incorrect identification was reinforced by confusing a *large* face for a *long* face (“*it's got quite a large face, it's not a small looking bumblebee, I think it's the garden bumblebee*”), finally deriving the wrong species identification. So, in the absence of direct experience in the field, the quality of decision-making using the Field guide was largely dependent upon the ability of a participant to recognise colour patterns and combine further attributes without ‘losing the plot’, thereby explaining why mean accuracy (Fig. 4A) was relatively high for easy images and very low for difficult ones.

### Decision tree

As with all decision trees, this interface had hierarchical and interface-directed interactive filtering, in our case from easy attributes to more difficult ones (i.e. from overall colour to colour patterns to distinguishing attributes), and with each feature decision on a separate page. Overall, interactive filtering led to a reduced workload as compared to the Field guide (Fig. 4C). This was particularly evident for colour patterns, with participants being able to readily limit choice, as for participant 3 (“. . . *can't see any yellow but not sure if that's due to the angle of the picture but I am gonna go with . . .* ”). Indeed, this

interface attracted the lowest perceived workload for difficult images (Fig. 4C), for which colour patterns could be rather complex.

Whilst this interface type facilitated high accuracy for difficult images, the Decision tree had the lowest accuracy for easier species, often because participants identified the distinguishing attributes incorrectly at the final stage. For example, participant 12 tried to match the remaining illustrations by colour ("... *because it's got quite bright orange on the thorax it could be moss carder bee*...") rather than using the distinguishing attributes as given on the final page. Similarly, participant 9 first appeared to have identified the specimen based on the distinguishing attribute ("... *rows of black hair on abdomen— I guess it's that*..."), but then gradually changed that decision by diverting attention to the illustrations ("... *ok these two require microscopic examination to separate*....*I do see black rows*...*I don't think its hairs really*...*if I had to choose*... *but that's a total guess from me*...") while continuously matching the image to illustrations, reinforcing the wrong choice and highlighting frustration with the choice eventually made.

In uncommon cases where remembering prior colour selections may have been less critical for making an accurate identification (which occurred only for one of the difficult images where the specimen was all black, eliminating the need to remember or match on the basis of colour patterns), participants were able to focus their attention on the distinguishing attribute (long vs. short face), as illustrated by participant 8 ("..*if I was feeding from a flower like that I think I would have a long face*..."), and participant 11 ("... *feel like its the longer face I'm going to go with this one yeah!!*...").

These observations highlight that even though participants were able to identify easier attributes based on colours and colour patterns correctly, and interactive elimination reduced their cognitive workload, the interface—in general—did not offer visual access to prior selections. So participants seemed to forget previously selected attributes and reverted back to pattern matching based on colours, rather than matching the distinguishing attributes given by the interface, which often led to incorrect identification. Further, while decisions were taken quickly for the colour patterns, participants dwelled on the final decision for a long time, often revisiting earlier colour choices in their minds, and this resulted in identifications taking as much time as on the other interfaces. So, while the Decision tree tool enabled participants to select and gain access to relevant information, even without holding direct experience of identification of bumblebees, the interface did not facilitate participants revisit their choices. The critical feed-back loop which sustains interactional expertise appeared thus to be missed out on.

### Feature selection

For easier images, using the Feature selection tool led to considerably higher accuracy than both the Field guide and even more so the Decision tree (Fig. 4A). The feature selection tool had an open choice interactive elimination, which helped participants to select clearer attributes first, for example, participant 16 ("... *thorax is entire black*... *the abdomen is orange*..."), as well as leave out attributes due to lack of confidence or occlusion, for example, participant 13 ("*wings*... *clear*... *pollen basket*... *not too sure*").

Additionally, the interface updated results from each selection, which helped participants to monitor their own progression along the identification path, as highlighted by participant 17 who first verified the results of previous selections ("...*leaves three species: one cuckoo, one rare and one common*..."), then selected a remaining attribute ("...*wings... look clear to me*...") to subsequently verify the results ("...*one rare... one common*..."). Such iterative interaction helped the participants to reduce to a small subset of similarly patterned species.

When participants seemed confused (due to image quality or colour variation) while identifying based on patterns, they could review their previous selections either by searching for other attribute options in drop-down menus, or by de-activating confusing attributes. For example, participant 11, who first selected an attribute (thorax) and said "...*potentially could be the tree bumblebee*..." but then de-activated this filter—as the colour from the image did not exactly match the colour in the drop-down list—and selected a different attribute ("...*abdomen is white and black*...") to get the same result (...*which again is the tree bumblebee*...). Such reviewing helped participants to verify prior selections by iteratively going back and forth for similarly patterned species, which facilitated their progression to identification steps concerning distinguishing attributes. For example, after verifying pattern selections for similar species, participant 4 progressed to identifying based on distinguishing attributes by reducing the number of candidate species from three to two ("...*never has black hairs so probably not that one*...") and then to one ("...*variable patches*..."), highlighting the Feature selection tool's ability to foster both progression and user confidence.

## DISCUSSION

This study looked at the role of digital interface design in citizens' engagement with species identification and, in particular, the learning processes which may sustain biological recording and associated scientific research as well as enhance public participation in respective citizen science initiatives (*Riesch & Potter, 2014*). A first important finding emerging from the study was the heavy reliance on visual classification as a process through which the human eye attends to specific features in order to make decisions about species identification. Notably, whilst this type of observation is common for field naturalists (*Ellis, 2011*), direct matching of illustrations may lead to errors by users with limited levels of expertise (*Austen et al., 2016*), as they may focus on irrelevant attributes perceived to be important for identification. Echoing prior research (*Wills, Inkster & Milton, 2015*), our results highlight this risk to be particularly significant with field guides, as these emphasise direct matching, a principle that leads to both increased cognitive workload and low accuracy of the identification task in novices.

So, while field guides may be the most common tools for identification, they prove difficult to use for novices and have the potential to lead to errors in the context of online species identification. The interactive elimination of choice reduced cognitive workload and helped participants to make progress during a classification process. This was evident for both Decision tree and Feature selection tools, where the interface helped participants to reduce their choice by selecting features. Both types of interactive

eliminations, directed and open choice, were useful for reducing the workload, as participants were able to limit choices to a subset of similarly patterned species. Cognitively, the option to select individual attributes is useful for identifying unknown objects (*Treisman & Gelade, 1980*; *Thompson & Massaro, 1989*; *Wills, Inkster & Milton, 2015*); such an option, allows an interface to direct user attention to specific features as well as reduce choices with each selection, and our results validate this for the task of bumblebee identification.

In the case of open choice, which was available in our Feature selection tool, the user progression may be easier for self-learning. As our findings also showed, when following a hierarchy of attributes, there is a risk for the identification going wrong due to overlooking or excluding particular attributes. This is notably true for novices, who may lack in confidence or sensitivity to particular features, but can also be the case for 'experts' who may be erring on the side of caution being aware of the level of complexity and intra-species variability (*Austen et al., 2016*). So, our findings seem to support the idea that open choice of filters may be more applicable for engaging users with different level of expertise, as they do not rely on a pre-defined hierarchy based on particular design assumptions about the user's experience or expertise (*Walter & Winterton, 2007*).

A general appraisal of our findings would thus suggest that in order for digital tools to fulfil their promise of increasing participation in citizen science projects (*Irwin, 1995*), it is important for novices and experts alike to be able to explore and learn about their own visual experiences. For this purpose, it may be important for all participants to iteratively attend to attributes for visual information processing in order to identify specimens correctly. This is even more relevant when platforms may be accessed from a wide geographical scale, calling for the identification of unfamiliar species. Even though all three interfaces had the provision for users to iteratively attend multiple attributes, Feature selection made this easiest (single page, with visual shading out of filtered options) and hence participants readily went back and forth during the identification process in order to verify possible mistakes and learn. Combining the advantages of interactive filtering and easier engagement, feature selection design seems to promote both user learning and species identification accuracy. Although our Decision tree design allowed participants to achieve similarly high accuracies for difficult images at a lower perceived workload than was the case using Feature selection, the latter interface proved far superior for easier images, and hence a better approach for classification tasks with variable levels of difficulty. To improve the decision tree design accuracies, human intelligence can be augmented with machine intelligence as in the case of mobile applications like the Merlin bird identification app where big data analytics is used to limit results by location and time of the year. However, such an approach requires reliable and fine-grained species distribution data to meaningfully reduce errors (*Farnsworth et al., 2013*)— a position that may not be within easy reach for many species groups other than birds (*Amano, Lamming & Sutherland, 2016*).

A range of documents unfold best practices and guidelines for the design of identification tools (*Walter & Winterton, 2007*; *Leggett & Kirchoff, 2011*; *Farnsworth et al., 2013*). Yet, we note, these are largely based on experiential knowledge, and very few

scientific studies exist which test elements of keys or field guides with users (*Stucky, 1984*; *Morse, Tardival & Spicer, 1996*; *Hawthorne, Cable & Marshall, 2014*; *Austen et al., 2016*). This problem is compounded by the exponential growth of citizen science projects incorporating identification tools, with multiple designs regularly present even within the same initiative (*Farnsworth et al., 2013*). Our findings may allow programmes to limit noise in citizen science data collected using embedded identification tools (see *Van der Wal et al., 2016* for an evaluation of our Feature selection tool embedded in a live citizen science programme), whilst potentially fostering the development of standardised identification technologies and learning. Greater awareness of design principles and learning processes in digital interfaces are thus essential for the identification of species groups heavily reliant on experts for identification—a resource known to be limited and diminishing (*Walter & Winterton, 2007*; *August et al., 2015*).

Species identification is a complex scientific task for which skills and expertise are gained through years of experience in the field as well as through training, either online or with other members of a community. Such learning requires cognitive support from identification tools and is thus an essential requirement for the new forms of online participation in community learning. Hence, there is a need to focus on user-centred design with emphasis on both usability and accuracy, which can engage new members in learning species identification skills while facilitating scientific research, both essential for citizen science. Returning to the framework of *Collins & Evans (2002)*, our research revealed that feature selection is a tool which allows non-experts to flexibly accrue relevant expert knowledge, enabling interactivity *as if* in a community of practice.

## CONCLUSION

The aim of the study presented was to understand how digital interfaces may be able to enhance decision-making and self-learning processes whilst participants are conducting a relatively complex task of species identification. The main findings are that open choice of filters were more user-friendly and improved user performance in the task as compared to interfaces which emphasised direct matching or hierarchical reasoning. Such findings have a number of important implications. First, by focussing on design principles fostering identification tasks using computing technologies, we sought to widen our perspective on citizen science. We moved from the original idea of citizens working as scientists (*Irwin, 1995*; *Riesch & Potter, 2014*) to a more flexible and hybrid model of citizens and future scientists operating and learning as part of extended communities of practice (*Lave & Wenger, 1991*). From this perspective, interdisciplinary dialogue across computing science, ecology and education was paramount to pave the way to more articulated processes of social learning centred on real-life problems (*Sol, Beers & Wals, 2013*). A second implication is that our findings run counter to the idea that citizens need to 'understand the science first'—as proposed by conventional models of public understanding of science (*Hage, Leroy & Petersen, 2010*)—in order to be able to partake in scientific research. Rather, our study points to the importance of developing awareness and making use of the very human process of sensorial perception in science (*Colucci-Gray & Camino, 2016*), both in order to learn and to design optimal interfaces.

As we have seen through the study and the findings, the possibility of iteratively attending to the attributes of a photographed specimen was found to be extremely valuable for learning and user-progression. These findings highlight that feature selection, which utilised these design principles effectively, may be one of the methods which can be used to productively integrate existing species identification knowledge, and allow novices to perform certain identification tasks readily and accurately. Finally, our findings provide support for the possibility of using digital tools to reinforce interactional as opposed to contributory expertise.

Given the prevalence of photo-submission-based species identification programmes, there is scope for extending the findings from this study to the design of interfaces that may be used in hybrid contexts of research and practice, such as in agro-ecology or urban gardening contexts. Following *Kahn, Severson & Ruckert (2009)*, our findings invite further research to fully embrace the potential for digital tools in citizen science projects to increase public's attention to environmental matters as well as greater engagement with and participation in science.

### Funding
This work was supported by the University of Aberdeen's Environment and Food Security Theme PhD studentship. The funders had no role in study design, data collection and analysis, decision to publish, or preparation of the manuscript.

### Grant Disclosures
The following grant information was disclosed by the authors:
University of Aberdeen's Environment and Food Security Theme PhD studentship.

### Competing Interests
The authors declare that they have no competing interests.

### Author Contributions
- Nirwan Sharma conceived and designed the experiments, performed the experiments, analysed the data, contributed reagents/materials/analysis tools, prepared figures and tables, authored and reviewed drafts of the paper, approved the final draft.
- Laura Colucci-Gray conceived and designed the experiments, contributed reagents/materials/analysis tools, authored and reviewed drafts of the paper, approved the final draft.
- Advaith Siddharthan conceived and designed the experiments, contributed reagents/materials/analysis tools, authored and reviewed drafts of the paper, approved the final draft.
- Richard Comont contributed reagents/materials/analysis tools, approved the final draft.
- René van der Wal conceived and designed the experiments, contributed reagents/materials/analysis tools, authored and reviewed drafts of the paper, approved the final draft.

## Human Ethics

The following information was supplied relating to ethical approvals (i.e. approving body and any reference numbers):

The study was approved by the University of Aberdeen's, College of Physical Sciences research ethics committee.

## Data Availability

The raw data is provided in the Supplementary File.

## Supplemental Information

Supplemental information for this article can be found online at http://dx.doi.org/10.7717/peerj.5965#supplemental-information.

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
