# Peer review of "Designing online species identification tools for biological recording: the impact on data quality and citizen science learning"

_PeerJ, doi:10.7717/peerj.5965_

## Round 0.1 · original submission · Minor Revisions

Having read the paper and the reviewers comments I am in agreement with their analysis that the manuscript just requires tightening up and clarifying in places.

·

Basic reporting

A well written paper which highlights the multi-faceted nature of citizen science projects. The authors provide a succinct and insightful overview of some of the challenges in contemporary biological recording, and discuss practical elements that are sometimes overlooked in other literature.

Experimental design

Pleased to see the use of mixed methods which offer an insight into the decision making process. The study could be readily replicated, or repeated with variation (e.g. different taxa or participant knowledge base).

Validity of the findings

I believe the findings to be of interest and useful to a range of people involved in biological recording. The authors have drawn on a variety of research areas, demonstrating that accurate species identification can depend upon a number of factors, and make suggestions on how to aid the learning process.

Additional comments

An interesting paper that addresses important aspects of species identification in the context of the wider use of technology. It is also encouraging to see further application of knowledge from other disciplines to species identification. A few comments:
• Line 62: Perhaps explain or reference the term ‘digital turn’? I’ve never come across the term and although I have found it in a web search, I wasn’t sure if it relates to a particular book or a wider concept that the book covers.
• Line 167: not important, but expanding Stevenson et al. to include Haber & Morris would show the reader that you are referring to the same paper being referenced throughout.
• Line 266: Should read Go Botany (not Botony).
• Line 318: reads “A paper version of the NASA-TLX subjective workload questionnaire (see below) was filled out by the participant for the completed identification task”. Does the see below refer to the explanation in lines 331-2? If so, would it be useful to include as an appendix?
• Line 343: should nasa read NASA?
• Lines 379, 394 & 416: which post hoc analyses were used?
• Participant consent (for the future): I would like to have seen another option for gender (e.g. other or prefer not to answer), especially as Aberdeen has an Athena Swan award.
• Perhaps one more careful read to identify double spaces or no spaces (e.g. before references in line 68 and 149), although this may just on my version of PDF (which has happened before!)

The paper was a pleasure to read, and just needs clarification in some places given that PeerJ is cross discipline and some readers may not be familiar with some of the concepts. Good luck!

·

Basic reporting

Parts of the literature review can be confusing, with sentences being too big and vague, not necessarily connected with the aim of the study.
Data classification applications in Citizen Science are very common, both in online and offline forms, but it is not mentioned or explained enough in the text.

Experimental design

Given that this is a usability based research for a citizen science application, trying to identify pain-points and the differences in learning for specific features, it could be useful to replicate the study with participants that are not familiar with any type of identification, and participants who are not involved with any type of environmental research.
Many of the users who use said applications are lay people, not necessarily familiar with any of the processes, which could potentially create different results in the study than the one with the conservation students.

Validity of the findings

no comment

Additional comments

Very useful research, covers areas in citizen science data identification that are not explored in-depth yet.

---

## Round 0.2 · accepted · Accept

Many thanks for the thoughtful response to the reviewers and amendments to the manuscript. I believe the manuscript has been significantly improved and happy to recommend it be accepted.

#